# KNOWLEDGE DISTILLATION BY SPARSE REPRESENTATION MATCHING

## ABSTRACT

Knowledge Distillation refers to a class of methods that transfers the knowledge from a teacher network to a student network. In this paper, we propose Sparse Representation Matching (SRM), a method to transfer intermediate knowledge obtained from one Convolutional Neural Network (CNN) to another by utilizing sparse representation learning. SRM first extracts sparse representations of the hidden features of the teacher CNN, which are then used to generate both pixel-level and image-level labels for training intermediate feature maps of the student network. We formulate SRM as a neural processing block, which can be efficiently optimized using stochastic gradient descent and integrated into any CNN in a plug-and-play manner. Our experiments demonstrate that SRM is robust to architectural differences between the teacher and student networks, and outperforms other KD techniques across several datasets.

## 1 INTRODUCTION

Over the past decade, deep neural networks have become the primary tools to tackle learning problems in several domains, ranging from machine vision (Ren et al., 2015; Redmon & Farhadi, 2018), natural language processing (Devlin et al., 2018; Radford et al., 2019) to biomedical analysis (Kiranyaz et al., 2015) or financial forecasting (Tran et al., 2018b; Zhang et al., 2019). Of those important developments, Convolutional Neural Networks have evolved as a de facto choice for high-dimensional signals, either as a feature extraction block or the main workhorse in a learning system. Initially developed in the 1990s for handwritten character recognition using only two convolutional layers (LeCun et al., 1998), state-of-the-art CNN topologies nowadays consist of hundreds of layers, having millions of parameters (Huang et al., 2017; Xie et al., 2017). In fact, not only in computer vision but also in other domains, state-of-the-art solutions are mainly driven by very large networks (Devlin et al., 2018; Radford et al., 2019), which limits their deployment in practice due to the high computational complexity.

The promising results obtained from maximally attainable computational power has encouraged a lot of research on developing smaller and light-weight models while achieving similar performances. This includes efforts on designing more efficient neural network families (both automatic and hand-crafted) (Howard et al., 2017; Tran et al., 2018a; 2019; Zoph & Le, 2016), compressing pretrained networks through weight pruning (Manessi et al., 2018; Tung & Mori, 2018), quantization (Hubara et al., 2017; Zhou et al., 2017) or approximation Denton et al. (2014); Jaderberg et al. (2014), as well as transferring knowledge from one network to another via knowledge distillation Hinton et al. (2015). Of these developments, Knowledge Distillation (KD) (Hinton et al., 2015) is a simple and widely used technique that has been shown to be effective in improving the performance of a network, given the access to one or many pretrained networks. KD and its variants work by utilizing the knowledge acquired in one or many models (the teacher(s)) as supervisory signals to train another model (the student) along with the labeled data. Thus, there are two central questions in KD:

- How to represent the knowledge encoded in a teacher network?
- How to efficiently transfer such knowledge to other networks, especially when there are architectural differences between the teacher and the student networks?

In the original formulation (Hinton et al., 2015), soft probabilities produced by the teacher represent its knowledge and the student network is trained to mimic this soft prediction. Besides the final

predictions, other works have proposed to utilize intermediate feature maps of the teacher as additional knowledge (Romero et al., 2014; Zagoruyko & Komodakis, 2016; Gao et al., 2018; Heo et al., 2019; Passalis & Tefas, 2018). Intuitively, intermediate feature maps contain certain clues on how the input is progressively transformed through layers of a CNN, thus can act as a good source of knowledge. However, we argue that the intermediate feature maps by themselves are not a good representation of the knowledge encoded in the teacher to teach the students. To address the question of representing the knowledge of the teacher CNN, instead of directly utilizing the intermediate feature maps of the teacher as supervisory signals, we propose to encode each pixel (of the feature maps) in a sparse domain and use the sparse representation as the source of supervision.

Prior to the era of deep learning, sparse representations attracted a great amount of interest in computer vision community and is a basis of many important works (Zhang et al., 2015). Sparse representation learning aims at representing the input signal in a domain where the coefficients are sparsest. This is achieved by using an overcomplete dictionary and decomposing the signal as a sparse linear combination of the atoms in the dictionary. While the dictionary can be prespecified, it is often desirable to optimize the dictionary together with the sparse decomposition using example signals. Since hidden feature maps in CNN are often smooth with high correlations between neighboring pixels, they are compressible, e.g., in Fourier domain. Thus, sparse representation serves as a good choice for representing information encoded in the hidden feature maps.

Sparse representation learning is a well-established topic in which several algorithms have been proposed (Zhang et al., 2015). However, to the best of our knowledge, existing formulations are computationally intensive to fit a large amount of data. Although learning task-specific sparse representations have been proposed in prior works (Mairal et al., 2011; Sprechmann et al., 2015; Monga et al., 2019), we have not seen its utilization for knowledge transfer using deep neural networks and stochastic optimization. In this work, we formulate sparse representation learning as a computation block that can be incorporated into any CNN and be efficiently optimized using mini-batch update from stochastic gradient-descent based algorithms. Our formulation allows us to take advantage of not only modern stochastic optimization techniques but also data augmentation to generate target sparsity on-the-fly.

Given the sparse representations obtained from the teacher network, we derive the target pixel-level and image-level sparse representation for the student network. Transferring knowledge from the teacher to the student is then conducted by optimizing the student with its own dictionaries to induce the target sparsity. Thus, our knowledge distillation method is dubbed as Sparse Representation Matching (SRM). Extensive experiments presented in Section 4 show that SRM significantly outperforms other recent KD methods, especially in transfer learning tasks by large margins. In addition, empirical results also indicate that SRM exhibits robustness to architectural mismatch between the teacher and the student.

## 2    RELATED WORK

The idea of transferring knowledge from one model to another has existed for a long time. This idea was first introduced in Breiman & Shang (1996) in which the authors proposed to grow decision trees to mimic the output of a complex predictor. Later, similar ideas were proposed for training neural networks (Ba & Caruana, 2014; Bucilu et al., 2006; Hinton et al., 2015), mainly for the purpose of model compression. Variants of the knowledge transfer idea differ in the methods of representing and transferring knowledge (Cao et al., 2018; Heo et al., 2019; Romero et al., 2014; Zagoruyko & Komodakis, 2016; Passalis & Tefas, 2018; Park et al., 2019; Tian et al., 2020), as well as the types of data being used (Lopes et al., 2017; Yoo et al., 2019; Papernot et al., 2016; Kimura et al., 2018).

In Bucilu et al. (2006), the final predictions of an ensemble on unlabeled data are used to train a single neural network. In Ba & Caruana (2014), the authors proposed to use the logits produced by the source network as the representation of knowledge, which is transferred to a target network by minimizing the Mean Squared Error (MSE) between the logits. The term Knowledge Distillation was introduced in Hinton et al. (2015) in which the student network is trained to simultaneously minimize the cross-entropy measured on the labeled data and the Kullback-Leibler (KL) divergence between its predicted probabilities and the soft probabilities produced by the teacher network. Since its introduction, this formulation has been widely adopted.

In addition to the soft probabilities of the teacher, later works have been proposed to utilize intermediate features of the teacher as additional sources of knowledge. For example, in FitNet (Romero et al., 2014), the authors referred to intermediate feature maps of the teacher as *hints* and the student is first pretrained by regressing its intermediate features to the teacher's hints, then later optimized with the standard KD approach. In other works, activation maps (Zagoruyko & Komodakis, 2016) as well as feature distributions (Passalis & Tefas, 2018) computed from intermediate layers have been proposed. In recent works (Park et al., 2019; Tian et al., 2020), instead of transferring knowledge about each individual sample, the authors proposed to transfer relational knowledge between pairs of samples.

Our SRM method bears some resemblances to previous works in the sense that SRM also uses intermediate feature maps as additional sources of knowledge. However, there are many differences between SRM and existing works. For example, in FitNet (Romero et al., 2014), the student network learns to regress from its intermediate features to the teacher's; however, the regressed features are not actually used in the student network. Thus, the hints in FitNet only *indirectly* influence the student's features. Since the sparse representation is another representation (equivalent) of the feature maps, SRM *directly* influences the student's features. In addition, by manipulating the sparse representation rather than the hidden features themselves, SRM is less prone to feature value range mismatch between the teacher and the student. This is because by construction, the sparse coefficients generated by SRM only have values in the range $[0, 1]$ as we will see in Section 3. Attention-based KD method (Zagoruyko & Komodakis, 2016) overcomes this problem by normalizing (using $l_2$ norm) the attention maps of the teacher and the student. This normalization step, however, might suffer from numerical instability (when $l_2$ norm is very small) when attention maps are calculated from activation layers such as ReLU.

In (Liu et al., 2019), the authors employed the idea of sparse coding, however, to represent the network's parameters rather than the intermediate feature maps as in our work. In (Jain et al., 2019), the authors used the idea of feature quantization and k-means clustering using intermediate features of the teacher to train the student with additional convolutional modules to predict the cluster labels. Our pixel-level label bears some resemblances to this method. However, we explicitly represent the intermediate features by sparse representation (by minimizing reconstruction error) and use the same process to transfer both local (pixel-level) and global (image-level) information.

## 3 Knowledge Distillation by Sparse Representation Matching

### 3.1 Knowledge Representation

Given the $n$-th input image $\mathcal{X}_n$, let us denote by $\mathcal{T}_n^{(l)} \in \mathbb{R}^{H_l \times W_l \times C_l}$ the output of the $l$-th layer of the teacher CNN, with $H_l$ and $W_l$ being the spatial dimensions and $C_l$ is the number of channels. In the following, we used the subscript $\mathcal{T}$ and $\mathcal{S}$ to denote a variable that is related to the teacher and student networks, respectively. In addition, we also denote by $\mathbf{t}_{n,i,j}^{(l)} = \mathcal{T}_n^{(l)}(i, j, :) \in \mathbb{R}^{C_l}$, which is the pixel at position $(i, j)$ of $\mathcal{T}_n^{(l)}$. The first objective in SRM is to represent each pixel $\mathbf{t}_{n,i,j}^{(l)}$ in a sparse domain. To do so, SRM learns an overcomplete dictionary of $M_l$ atoms: $\mathbf{D}_{\mathcal{T}}^{(l)} = [\mathbf{d}_{\mathcal{T},1}^{(l)}, \ldots, \mathbf{d}_{\mathcal{T},M_l}^{(l)}] \in \mathbb{R}^{C_l \times M_l}$ ($M_l > C_l$), which is used to express each pixel $\mathbf{t}_{n,i,j}^{(l)}$ as a linear combination of $\mathbf{d}_{\mathcal{T},m}^{(l)}$ as follows:

$$\mathbf{t}_{n,i,j}^{(l)} = \sum_{m=1}^{M_l} \psi_k(\mathbf{t}_{n,i,j}^{(l)}, \mathbf{d}_{\mathcal{T},m}^{(l)}) \cdot \kappa(\mathbf{t}_{n,i,j}^{(l)}, \mathbf{d}_{\mathcal{T},m}^{(l)}) \cdot \mathbf{d}_{\mathcal{T},m}^{(l)} \tag{1}$$

where

- $\kappa(\mathbf{t}_{n,i,j}^{(l)}, \mathbf{d}_{\mathcal{T},m}^{(l)})$ denotes a function that measures the similarity between $\mathbf{t}_{n,i,j}^{(l)}$ and atom $\mathbf{d}_{\mathcal{T},m}^{(l)}$. We further denote by $\mathbf{k}_{\mathcal{T},n,i,j}^{(l)} = [\kappa(\mathbf{t}_{n,i,j}^{(l)}, \mathbf{d}_{\mathcal{T},1}^{(l)}), \ldots, \kappa(\mathbf{t}_{n,i,j}^{(l)}, \mathbf{d}_{\mathcal{T},M_l}^{(l)})]$ the vector that contains similarities between $\mathbf{t}_{n,i,j}^{(l)}$ and all atoms in the dictionary $\mathbf{D}_{\mathcal{T}}^{(l)}$.

- $\psi_k(\mathbf{t}_{n,i,j}^{(l)}, \mathbf{d}_m^{(l)})$ denotes the indicator function that returns a value of 1 if $\kappa(\mathbf{t}_{n,i,j}^{(l)}, \mathbf{d}_{\mathcal{T},m}^{(l)})$ belongs to the set of top-$k$ values in $\mathbf{k}_{\mathcal{T},n,i,j}^{(l)}$, and a value of 0 otherwise.

The decomposition in Eq. (1) basically means that $\mathbf{t}_{n,i,j}^{(l)}$ is expressed as the linear combination of $k$ most similar atoms in $\mathbf{D}_{\mathcal{T}}^{(l)}$, with the coefficients being the corresponding similarity values. Let $\lambda_{n,i,j,m}^{(l)} = \psi_k(\mathbf{t}_{n,i,j}^{(l)}, \mathbf{d}_{\mathcal{T},m}^{(l)}) \cdot \kappa(\mathbf{t}_{n,i,j}^{(l)}, \mathbf{d}_{\mathcal{T},m}^{(l)})$, then the sparse representation of $\mathbf{t}_{n,i,j}^{(l)}$ is then defined as:

$$\tilde{\mathbf{t}}_{n,i,j}^{(l)} = [\lambda_{n,i,j,1}^{(l)}, \ldots, \lambda_{n,i,j,M_l}^{(l)}] \in \mathbb{R}^{M_l} \tag{2}$$

By construction, there are only $k$ non-zero elements in $\tilde{\mathbf{t}}_{n,i,j}^{(l)}$, and $k$ defines the degree of sparsity, which is a hyper-parameter of SRM. In order to find $\tilde{\mathbf{t}}_{n,i,j}^{(l)}$, we simply minimize the reconstruction error in Eq. (1) as follows:

$$\underset{\mathbf{D}_{\mathcal{T}}^{(l)}}{\arg\min} \sum_{n,i,j} \left\| \mathbf{t}_{n,i,j}^{(l)} - \sum_{m=1}^{M_l} \psi_k(\mathbf{t}_{n,i,j}^{(l)}, \mathbf{d}_{\mathcal{T},m}^{(l)}) \cdot \kappa(\mathbf{t}_{n,i,j}^{(l)}, \mathbf{d}_{\mathcal{T},m}^{(l)}) \cdot \mathbf{d}_{\mathcal{T},m}^{(l)} \right\|_2^2 \tag{3}$$

There are many choices for the similarity function $\kappa$ such as linear kernel, RBF kernel, sigmoid kernel and so on. In our work, we used the sigmoid kernel $\kappa(\mathbf{x}, \mathbf{y}) = sigmoid(\mathbf{x}^T \mathbf{y} + c)$ since the dot-product makes it computationally efficient and the gradients in the backward pass are stable. Although the RBF kernel is popular in many works, we empirically found that RBF kernel is sensitive to the learning rate, which easily leads to numerical issues.

## 3.2 Transferring Knowledge

Let us denote by $\mathcal{S}_n^{(p)} \in \mathbb{R}^{H_p \times W_p \times C_p}$ the output of the $p$-th layer of the student network given input image is $\mathcal{X}_n$. In addition, $\mathbf{s}_{n,i,j}^{(p)} \in \mathbb{R}^{C_p}$ denotes the pixel at position $(i, j)$ of $\mathcal{S}_n^{(p)}$. We consider the task of transferring knowledge from the $l$-th layer of the teacher to the $p$-th layer of the student. To do so, we require that the spatial dimensions of both networks match ($H_p = H_l$ and $W_p = W_l$) while the channel dimensions might differ.

Given the sparse representation of the teacher in Eq. (2), a straightforward way is to train the student network to produce hidden features at spatial position $(i, j)$, having the same sparse coefficients as its teacher. However, trying to learn exact sparse representations as produced by the teacher is a too restrictive task since this enforces learning the absolute value of every point in a high-dimensional space. Instead of enforcing an absolute constraint on how each pixel of every sample should be represented, to transfer knowledge, we only enforce a relative constraints between them in the sparse domain. Specifically, we propose to train the student to only approximate sparse structures of the teacher network by solving a classification problem with two types of labels extracted from the sparse representation $\tilde{\mathbf{t}}_{n,i,j}^{(l)}$ of the teacher: pixel-level and image-level labels.

**Pixel-level labeling**: for each spatial position $(i, j)$, we assign a class label, which is the index of the largest element of $\tilde{\mathbf{t}}_{n,i,j}^{(l)}$, i.e., the index of the closest (most similar) atom in $\mathbf{D}_{\mathcal{T}}^{(l)}$. This basically means that we partition all pixels into $M_l$ disjoint sets using dictionary $\mathbf{D}_{\mathcal{T}}^{(l)}$, and the student network is trained to learn the same partitioning using its own dictionary $\mathbf{D}_{\mathcal{S}}^{(p)} = [\mathbf{d}_{\mathcal{S},1}^{(p)}, \ldots, \mathbf{d}_{\mathcal{S},M_l}^{(p)}] \in \mathbb{R}^{C_p \times M_l}$. Let $\mathbf{k}_{\mathcal{S},n,i,j}^{(p)} = [\kappa(\mathbf{s}_{n,i,j}^{(p)}, \mathbf{d}_{\mathcal{S},1}^{(p)}), \ldots, \kappa(\mathbf{s}_{n,i,j}^{(p)}, \mathbf{d}_{\mathcal{S},M_l}^{(p)})]$ denote the vector that contains similarities between pixel $\mathbf{s}_{n,i,j}^{(p)}$ and $M_l$ atoms in $\mathbf{D}_{\mathcal{S}}^{(p)}$. The first knowledge transfer objective in our method using pixel-level label is defined as follows:

$$\underset{\mathbf{\Theta}_{\mathcal{S}}, \mathbf{D}_{\mathcal{S}}^{(p)}}{\arg\min} \sum_{n,i,j} \mathcal{L}_{CE}(c_{n,i,j}, \mathbf{k}_{\mathcal{S},n,i,j}^{(p)}) \tag{4}$$

where $\mathbf{\Theta}_{\mathcal{S}}$ denotes parameters of the student network. $\mathcal{L}_{CE}$ denotes the cross-entropy loss function, and $c_{n,i,j} = \arg\max(\tilde{\mathbf{t}}_{n,i,j}^{(l)})$. Here we should note that the idea of transferring the structure instead of the absolute representation is not new. For example, in (Park et al., 2019) and (Tian et al., 2020), the authors proposed to transfer the relative distance between the embeddings of samples. In our case, the pixel-level labels provide supervisory information on how the pixels in the student network should be represented in the sparse domain so that their partition using the nearest atom is the same.

**Image-level labeling**: given the sparse representation $\tilde{\mathbf{t}}_{n,i,j}^{(l)}$ of the teacher, we generate an image-level label by averaging $\tilde{\mathbf{t}}_{n,i,j}^{(l)}$ over the spatial dimensions. While pixel-level labels provide local

Table 1: Performance in CIFAR100

| Model | Test Accuracy % |
|---|---|
| DenseNet121 (teacher) | $75.09 \pm 00.29$ |
| AllCNN | $67.64 \pm 01.87$ |
| AllCNN-KD | $73.27 \pm 00.20$ |
| AllCNN-FitNet | $72.03 \pm 00.27$ |
| AllCNN-AT | $70.88 \pm 0.29$ |
| AllCNN-PKT | $72.22 \pm 0.35$ |
| AllCNN-RKD | $70.39 \pm 0.17$ |
| AllCNN-CRD | $72.70 \pm 0.18$ |
| AllCNN-SRM (our) | $\mathbf{74.73} \pm 00.26$ |

supervisory information encoding the spatial information, image-level label provides global supervisory information, promoting the shift-invariance property. Image-level label bears some resemblances to the Bag-of-Feature model (Passalis & Tefas, 2017), which aggregates the histograms of image patches to generate an image-level feature. The second knowledge transfer objective in our method using image-level labels is defined as follows:

$$\underset{\boldsymbol{\Theta}_{\mathcal{S}}, \mathbf{D}_{\mathcal{S}}^{(p)}}{\arg\min} \sum_{n} \mathcal{L}_{BCE}\left(\frac{\sum_{i,j} \tilde{\mathbf{t}}_{n,i,j}^{(l)}}{H_l \cdot W_l}, \frac{\sum_{i,j} \mathbf{k}_{\mathcal{S},n,i,j}^{(p)}}{H_l \cdot W_l}\right) \tag{5}$$

where $\mathcal{L}_{BCE}$ denotes the binary cross-entropy loss. Here we should note that since most kernel functions output a similarity score in $[0, 1]$, elements of $\tilde{\mathbf{t}}_{n,i,j}^{(l)}$ and $\mathbf{k}_{\mathcal{S},n,i,j}^{(p)}$ are also in this range, making the two inputs to $\mathcal{L}_{BCE}$ in Eq. (5) valid.

To summarize, the procedures of our SRM method is similar to FitNet (Romero et al., 2014), which consists of the following steps:

- **Step 1**: Given the source layers (with indices $l$) in the teacher network $\mathcal{T}$, find the sparse representation by solving Eq. (3).
- **Step 2**: Given the target layers (with indices $p$), optimize the student network $\mathcal{S}$ to predict pixel-level and image-level labels by solving Eq. (4), (5).
- **Step 3**: Given the student network obtained in Step 2, optimize it using the original KD algorithm.

All optimization objectives in our algorithm are solved by stochastic gradient descent.

## 4 EXPERIMENTS

The first set of experiments was conducted on CIFAR100 dataset (Krizhevsky et al., 2009) to compare our SRM method with other KD methods: KD (Hinton et al., 2015), FitNet (Romero et al., 2014), AT (Zagoruyko & Komodakis, 2016), PKT (Passalis & Tefas, 2018), RKD (Park et al., 2019) and CRD (Tian et al., 2020). In the second set of experiments, we tested the algorithms under transfer learning setting. In the final set of experiments, we evaluated SRM on the large-scale ImageNet dataset.

For every experiment configuration, we ran 3 times and reported the mean and standard deviation. Regarding the source and the target layers for transferring intermediate knowledge, we simply selected the outputs of the downsampling layers. Detailed information about our experimental setup is provided in the Appendices.

### 4.1 EXPERIMENTS ON CIFAR100

Since CIFAR100 has no validation set, we randomly selected 5K samples from the training set for validation purpose, reducing the training set size to 45K. Our setup is different from the conventional practice of validating and reporting the result on the test set of CIAR100. In this set of experiments, we

Table 2: Comparison (test accuracy %) between FitNet and SRM in terms of quality of intermediate knowledge transfer on CIFAR100

|  | AllCNN-FitNet | AllCNN-SRM (our) |
|---|---|---|
| Linear Probing | $\mathbf{69.95} \pm 00.20$ | $69.10 \pm 00.31$ |
| Whole Network Update | $68.06 \pm 00.10$ | $\mathbf{71.99} \pm 00.08$ |

used DenseNet121 (Huang et al., 2017) as the teacher network (7.1M parameters and 900M FLOPs) and a non-residual architecture (a variant of AllCNN network (Springenberg et al., 2014)) as the student network (2.2M parameters and 161M FLOPs). Other details about training hyperparameters can be found in our Appendix A.

**Overall comparison** (Table 1): It is clear that the student network significantly benefits from all knowledge transfer methods. The proposed SRM method clearly outperforms other competing methods, establishing more than $1\%$ margin compared to the second best method (KD). In fact, the student network trained with SRM achieves very close performance with its teacher ($74.73\%$ versus $75.09\%$), despite having a non-residual architecture and $3\times$ less parameters. In addition, recent methods, such as PKT and CRD, perform better than FitNet, however, inferior to the original KD method.

**Quality of intermediate knowledge transfer**: Since FitNet and SRM have a pretraining phase to transfer intermediate knowledge, we compared the quality of intermediate knowledge transferred by FitNet and SRM by conducting two types of experiments after transferring intermediate knowledge: (1) the student network is optimized for 60 epochs using only the training data (without teacher's soft probabilities), given all parameters are fixed, except the last linear layer for classification. This experiment is called *Linear Probing*; (2) the student network is optimized with all parameters for 200 epochs using only the training data (without teacher's soft probabilities). This experiment is named *Whole Network Update*. The results are shown in Table 2.

Firstly, both experiments show that the student networks outperform their baseline, thus, benefit from intermediate knowledge transfer by both methods, even without the final KD phase. In the first experiment when we only updated the output layer, the student pretrained by FitNet achieves slightly better performance than by SRM ($69.95\%$ versus $69.10\%$). However, when we optimized with respect to all parameters, the student pretrained by SRM performs significantly better than the one pretrained by FitNet ($71.99\%$ versus $68.06\%$). While full parameter update led the student pretrained by SRM to better local optima, it undermines the student pretrained by FitNet. This result suggests that the process of intermediate knowledge transfer in SRM can initialize the student network at better positions in the parameter space compared to FitNet.

**Effects of sparsity ($\lambda$) and dictionary size ($\mu$)**: in Table 3, we show the performance of SRM with different degrees of sparsity (parameterized by $\lambda = k/M_l$, lower $\lambda$ indicates higher sparsity) and dictionary sizes (parameterized by $\mu = M_l/C_l$, higher $\mu$ indicates higher overcompleteness). As can be seen from Table 3, SRM is not sensitive to $\lambda$ and $\mu$. In fact, the worst configuration still performs slightly better than KD ($73.71\%$ versus $73.27\%$), and much better than other AT, PKT, RKD and CRD.

Table 3: SRM test accuracy (%) on CIFAR100 with different dictionary sizes (parameterized by $\mu$) and degree of sparsity (parameterized by $\lambda$)

|  | $\mu = 1.5$ | $\mu = 2.0$ | $\mu = 3.0$ |
|---|---|---|---|
| $\lambda = 0.01$ | $74.00 \pm 00.15$ | $74.12 \pm 00.35$ | $74.09 \pm 00.08$ |
| $\lambda = 0.02$ | $74.34 \pm 00.07$ | $74.73 \pm 00.26$ | $74.20 \pm 00.27$ |
| $\lambda = 0.03$ | $73.77 \pm 00.05$ | $73.83 \pm 00.12$ | $73.71 \pm 00.51$ |

**Pixel-level label and image-level label**: finally, to show the importance of combining both pixel-level and image-level label, we experimented with two other variants of SRM on CIFAR100: using either pixel-level or image-level label. The results are shown in Table 4. The student network

performs poorly when only receiving intermediate knowledge via image-level labels, even though it was later optimized with the standard KD phase. Similar to the observations made from Table 2, this again suggests that the position in the parameter space, which is obtained after the intermediate knowledge transfer phase, and prior to the standard KD phase, heavily affects the final performance. Once the student network is badly initialized, even receiving soft probabilities from the teacher as additional signals does not help. Although using pixel-level label alone is better than image-level label, the best performance is obtained by combining both objectives.

Table 4: Effects of pixel-level label and image-level label in SRM on CIFAR100

| Pixel-level label | Image-level label | Test accuracy % |
| --- | --- | --- |
| ✓ | | $73.16 \pm 00.39$ |
| | ✓ | $53.50 \pm 09.96$ |
| ✓ | ✓ | $\mathbf{74.73} \pm 00.26$ |

## 4.2 Transfer Learning Experiments

Since transfer learning is key in the success of many applications that utilize deep neural networks, we conducted experiments in 5 transfer learning problems (Flowers (Nilsback & Zisserman, 2008), CUB (Wah et al., 2011), Cars (Krause et al., 2013), Indoor-Scenes (Quattoni & Torralba, 2009) and PubFig83 (Pinto et al., 2011)) to assess how well the proposed method works under transfer learning setting compared to others.

In our experiments, we used a pretrained ResNext50 (Xie et al., 2017) on ILSVRC2012 database as the teacher network, which is finetuned using the training set of each transfer learning task. We then benchmarked how well each knowledge distillation method transfers both pretrained knowledge and domain specific knowledge to a *randomly initialized* student network using domain specific data. Both residual (ResNet18 (He et al., 2016), 11.3M parameters, 1.82G FLOPs) and non-residual (a variant of AllCNN (Springenberg et al., 2014), 5.1M parameters, 1.35G FLOPs) architectures were used as the student.

**Full transfer setting**: in this setting, we used all samples available in the training set to perform knowledge transfer from the teacher to the student. The test accuracy achieved by different methods is shown in the upper part of Table 5. It is clear that the proposed SRM outperforms other methods on many datasets, except on Cars and Indoor-Scenes datasets for AllCNN student. While KD, AT, PKT, CRD and SRM successfully led the students to better minima with both residual or non-residual students, FitNet was only effective with the residual one. The results suggest that the proposed intermediate knowledge transfer mechanism in SRM is robust to architectural differences between the teacher and the student networks.

**Few-shot transfer setting**: we further assessed how well the methods perform when there is a limited amount of data for knowledge transfer. For each dataset, we randomly selected 5 samples (5-shot) and 10 samples (10-shot) from the training set for training purpose, and kept the validation and test set similar to the full transfer setting. Since the Flowers dataset has only 10 training samples in total (the original split provided by the database has 20 samples per class, however, we used 10 samples for validation purpose), the results for 10-shot are similar to full transfer learning setting. The test performance (% in accuracy) is reported in the lower part of Table 5. Under this restrictive regime, the proposed SRM method performs far better than other tested methods for both types of students, largely improves the baseline results.

## 4.3 ImageNet Experiments

For ImageNet (Russakovsky et al., 2015) experiments, we followed similar experimental setup as in (Cho & Hariharan, 2019; Zagoruyko & Komodakis, 2016): ResNet34 and ResNet18 were used as the teacher and student networks, respectively. Table 6 shows top-1 classification errors of SRM and related methods having the same experimental setup. The teacher network (Resnet34) achieves top-1 error of 26.70%. Using SRM, we can successfully train ResNet18 to achieve 28.79% classification error, which is lower than existing methods (except CRD), including the recently proposed ESKD+AT

Table 5: Transfer learning using AllCNN (ACNN) and ResNet18 (RN18) (test accuracy %). The standard deviation of the test accuracy in few-shot settings is reported in Table 7

| Model | Flowers | CUB | Cars | Indoor-Scenes | PubFig83 |
|---|---|---|---|---|---|
| ResNext50 | $89.35 \pm 00.62$ | $69.53 \pm 00.45$ | $87.45 \pm 00.27$ | $63.51 \pm 00.43$ | $91.41 \pm 00.14$ |
| **Full Shot** | | | | | |
| ACNN | $40.80 \pm 02.33$ | $47.26 \pm 00.18$ | $61.93 \pm 01.38$ | $35.82 \pm 00.43$ | $78.47 \pm 00.17$ |
| ACNN-KD | $46.14 \pm 00.39$ | $51.80 \pm 00.41$ | $66.12 \pm 00.17$ | $38.44 \pm 00.99$ | $81.54 \pm 00.09$ |
| ACNN-FitNet | $30.10 \pm 02.41$ | $44.30 \pm 01.25$ | $60.20 \pm 03.83$ | $30.87 \pm 00.35$ | $77.61 \pm 00.87$ |
| ACNN-AT | $51.62 \pm 00.69$ | $51.74 \pm 00.39$ | $\mathbf{73.89} \pm 00.06$ | $\mathbf{43.56} \pm 00.52$ | $81.11 \pm 01.51$ |
| ACNN-PKT | $47.12 \pm 00.52$ | $47.60 \pm 00.85$ | $70.16 \pm 00.51$ | $37.71 \pm 00.64$ | $82.03 \pm 00.26$ |
| ACNN-RKD | $42.00 \pm 01.10$ | $39.99 \pm 00.61$ | $56.99 \pm 02.44$ | $30.94 \pm 00.45$ | $75.44 \pm 00.50$ |
| ACNN-CRD | $46.99 \pm 01.13$ | $51.12 \pm 00.34$ | $68.89 \pm 00.47$ | $42.82 \pm 00.21$ | $\mathbf{83.02} \pm 00.11$ |
| ACNN-SRM | $\mathbf{51.72} \pm 00.58$ | $\mathbf{54.51} \pm 01.72$ | $71.44 \pm 04.76$ | $43.09 \pm 00.60$ | $82.89 \pm 01.78$ |
| RN18 | $44.25 \pm 00.42$ | $44.79 \pm 00.68$ | $57.17 \pm 01.95$ | $36.72 \pm 00.29$ | $79.08 \pm 00.36$ |
| RN18-KD | $48.26 \pm 00.33$ | $54.91 \pm 00.33$ | $75.29 \pm 00.46$ | $43.84 \pm 00.67$ | $84.49 \pm 00.28$ |
| RN18-FitNet | $48.29 \pm 01.24$ | $61.28 \pm 00.48$ | $\mathbf{85.01} \pm 00.10$ | $45.93 \pm 01.00$ | $89.78 \pm 00.20$ |
| RN18-AT | $51.49 \pm 00.42$ | $53.13 \pm 00.40$ | $77.14 \pm 00.15$ | $44.13 \pm 00.55$ | $83.60 \pm 00.19$ |
| RN18-PKT | $45.32 \pm 00.51$ | $45.24 \pm 00.39$ | $71.24 \pm 02.23$ | $37.27 \pm 00.79$ | $82.00 \pm 00.10$ |
| RN18-RKD | $42.32 \pm 00.28$ | $36.29 \pm 00.58$ | $56.87 \pm 02.64$ | $29.57 \pm 00.90$ | $70.90 \pm 01.79$ |
| RN18-CRD | $47.67 \pm 00.05$ | $53.25 \pm 00.83$ | $76.51 \pm 00.74$ | $43.76 \pm 00.40$ | $84.43 \pm 00.40$ |
| RN18-SRM | $\mathbf{67.46} \pm 01.06$ | $\mathbf{63.11} \pm 00.45$ | $84.40 \pm 00.67$ | $\mathbf{54.59} \pm 00.38$ | $\mathbf{89.79} \pm 00.53$ |
| **5-Shot → 10-shot** | | | | | |
| ACNN | $32.91 \rightarrow 40.80$ | $13.19 \rightarrow 25.53$ | $05.03 \rightarrow 09.50$ | $09.21 \rightarrow 16.23$ | $05.15 \rightarrow 10.09$ |
| ACNN-KD | $35.98 \rightarrow 46.14$ | $21.60 \rightarrow 34.63$ | $10.61 \rightarrow 23.43$ | $14.81 \rightarrow 21.76$ | $11.97 \rightarrow 28.11$ |
| ACNN-FitNet | $28.73 \rightarrow 30.10$ | $14.78 \rightarrow 29.40$ | $06.11 \rightarrow 15.81$ | $08.36 \rightarrow 15.71$ | $08.25 \rightarrow 17.89$ |
| ACNN-AT | $38.21 \rightarrow 51.62$ | $17.69 \rightarrow 30.26$ | $08.52 \rightarrow 25.22$ | $08.76 \rightarrow 17.90$ | $08.02 \rightarrow 26.27$ |
| ACNN-PKT | $33.25 \rightarrow 47.12$ | $11.30 \rightarrow 24.93$ | $06.16 \rightarrow 13.82$ | $10.75 \rightarrow 16.78$ | $06.13 \rightarrow 10.67$ |
| ACNN-RKD | $30.27 \rightarrow 42.00$ | $09.55 \rightarrow 19.36$ | $04.82 \rightarrow 09.60$ | $09.68 \rightarrow 12.40$ | $04.82 \rightarrow 06.99$ |
| ACNN-CRD | $35.01 \rightarrow 46.99$ | $18.09 \rightarrow 29.72$ | $06.77 \rightarrow 16.96$ | $09.73 \rightarrow 17.60$ | $06.33 \rightarrow 17.24$ |
| ACNN-SRM | $\mathbf{41.14} \rightarrow \mathbf{51.72}$ | $\mathbf{22.89} \rightarrow \mathbf{35.86}$ | $\mathbf{11.71} \rightarrow \mathbf{36.28}$ | $\mathbf{16.63} \rightarrow \mathbf{24.82}$ | $\mathbf{13.96} \rightarrow \mathbf{31.47}$ |
| RN18 | $32.95 \rightarrow 44.25$ | $11.55 \rightarrow 22.72$ | $05.00 \rightarrow 11.76$ | $09.04 \rightarrow 15.16$ | $04.98 \rightarrow 08.92$ |
| RN18-KD | $38.07 \rightarrow 48.26$ | $25.53 \rightarrow 40.57$ | $11.37 \rightarrow 35.44$ | $14.61 \rightarrow 23.85$ | $10.69 \rightarrow 29.67$ |
| RN18-FitNet | $39.17 \rightarrow 48.29$ | $26.50 \rightarrow 43.83$ | $12.36 \rightarrow 51.88$ | $13.72 \rightarrow 24.62$ | $11.49 \rightarrow 36.79$ |
| RN18-AT | $37.36 \rightarrow 51.49$ | $18.22 \rightarrow 30.47$ | $08.96 \rightarrow 27.70$ | $09.93 \rightarrow 17.48$ | $08.76 \rightarrow 29.80$ |
| RN18-PKT | $33.24 \rightarrow 45.32$ | $10.63 \rightarrow 20.62$ | $05.88 \rightarrow 11.00$ | $11.03 \rightarrow 15.76$ | $05.34 \rightarrow 08.80$ |
| RN18-RKD | $29.97 \rightarrow 42.32$ | $08.83 \rightarrow 17.73$ | $04.98 \rightarrow 08.73$ | $08.41 \rightarrow 11.82$ | $04.49 \rightarrow 06.49$ |
| RN18-CRD | $34.24 \rightarrow 47.67$ | $18.36 \rightarrow 30.04$ | $06.95 \rightarrow 17.11$ | $09.01 \rightarrow 17.82$ | $06.36 \rightarrow 16.08$ |
| RN18-SRM | $\mathbf{51.24} \rightarrow \mathbf{67.46}$ | $\mathbf{34.67} \rightarrow \mathbf{48.42}$ | $\mathbf{26.63} \rightarrow \mathbf{61.22}$ | $\mathbf{21.18} \rightarrow \mathbf{32.21}$ | $\mathbf{29.31} \rightarrow \mathbf{51.99}$ |

(Cho & Hariharan, 2019) that combines early-stopping trick and Attention Transfer (Zagoruyko & Komodakis, 2016).

Table 6: Top-1 Error of ResNet18 on ImageNet. (*) indicates results obtained by 110 epochs. References of the methods in this table are mentioned in the Appendix C

| KD | AT | Seq. KD | KD+ONE | ESKD | ESKD+AT | CRD* | SRM (our) |
|------|------|------|------|------|------|------|------|
| 30.79 | 29.30 | 29.60 | 29.45 | 29.16 | 28.84 | 28.62* | 28.79 |

## 5 CONCLUSION

In this work, we proposed Sparse Representation Matching (SRM), a method to transfer intermediate knowledge from one network to another using sparse representation learning. Experimental results on several datasets indicated that SRM outperforms related methods, successfully performing intermediate knowledge transfer even if there is a significant architectural mismatch between networks and/or a limited amount of data. SRM serves as a starting point for developing specific knowledge transfer use cases, e.g., data-free knowledge transfer, which is an interesting future research direction.

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

## A    CIFAR100

In this experiment, we used ADAM optimizer and trained all networks for 200 epochs with the initial learning rate of $0.001$, which is reduced by $0.1$ at epochs 31 and 131. For those methods that have pre-training phase, the number of epochs for pre-training was set to 160. For regularization, we used weight decay with coefficient $0.0001$. For data augmentation, we followed the conventional protocol, which randomly performs horizontal flipping, random horizontal and/or vertical shifting by four pixels. For SRM, KD and FitNet, we used our own implmentation, for other methods (AT, PKT, RKD, CRD), we used the code provided by the authors of CRD method (Tian et al., 2020).

For KD, FitNet and SRM, we validated the temperature $\tau$ (used to soften teacher's probability) and the weight $\alpha$ (used to balance between classification and distillation loss) from the following set: $\tau \in \{2.0, 4.0, 8.0\}$, and $\alpha \in \{0.25, 0.5, 0.75\}$. For other methods, we used the provided values used in CRD paper (Tian et al., 2020). In addition, there are two hyperparameters of SRM: the degree of sparsity ($\lambda = k/M_l$) and overcompleteness of dictionaries ($\mu = M_l/C_l$). Lower values of $\lambda$ indicate sparser representations while higher values of $\mu$ indicate larger dictionaries. For CIFAR100, we performed validation with $\lambda \in \{0.01, 0.02, 0.03\}$ and $\mu = \{1.5, 2.0, 3.0\}$.

## B    TRANSFER LEARNING

For each transfer learning dataset, we randomly sampled a few samples from the training set to establish the validation set. All images were resized to resolution $224 \times 224$ and we used standard ImageNet data augmentation (random crop and horizontal flipping) during stochastic optimization. Based on the analysis of hyperparameters in CIFAR100 experiments, we validated KD, FitNet and SRM using $\alpha \in \{0.5, 0.75\}$ and $\tau \in \{4.0, 8.0\}$. Sparsity degree $\lambda = 0.02$ and dictionary size $\mu = 2.0$ were used for SRM. For other methods, we used the hyperparameter settings provided in CRD paper (Tian et al., 2020). All experiments were conducted using ADAM optimizer for 200 epochs with the initial learning rate of $0.001$, which is reduced by $0.1$ at epochs 41 and 161. The weight decay coefficient was set to $0.0001$.

## C    IMAGENET

For hyperparameters of SRM, we set $\mu = 4.0, \lambda = 0.02, \tau = 4.0, \alpha = 0.3$. For other training protocols, we followed the standard setup for ImageNet, which trains the student network for 100 epochs using SGD optimizer with an initial learning rate of $0.1$, dropping by $0.1$ at epochs 51, 81, 91.

Table 7: Transfer learning: standard deviation (%) of test accuracy

| Model | Flowers | CUB | Cars | Indoor-Scenes | PubFig83 |
|---|---|---|---|---|---|
| 5-Shot | | | | | |
| ACNN | 00.94 | 00.46 | 00.11 | 00.81 | 00.45 |
| ACNN-KD | 00.76 | 00.83 | 00.52 | 00.67 | 01.40 |
| ACNN-FitNet | 01.18 | 00.60 | 00.37 | 00.52 | 01.36 |
| ACNN-AT | 00.88 | 00.94 | 01.50 | 00.39 | 00.70 |
| ACNN-PKT | 00.31 | 00.20 | 00.29 | 01.04 | 00.18 |
| ACNN-RKD | 00.27 | 00.34 | 00.31 | 00.39 | 00.16 |
| ACNN-CRD | 00.57 | 00.91 | 00.55 | 00.49 | 00.28 |
| ACNN-SRM | 01.09 | 01.18 | 01.21 | 00.34 | 00.52 |
| RN18 | 00.37 | 00.10 | 00.27 | 00.44 | 00.22 |
| RN18-KD | 01.47 | 00.35 | 00.58 | 00.88 | 00.99 |
| RN18-FitNet | 00.62 | 00.46 | 01.00 | 01.07 | 00.66 |
| RN18-AT | 01.23 | 00.47 | 00.92 | 00.73 | 00.34 |
| RN18-PKT | 00.47 | 00.18 | 00.18 | 00.82 | 00.11 |
| RN18-RKD | 00.55 | 00.40 | 00.13 | 00.49 | 00.24 |
| RN18-CRD | 00.32 | 02.09 | 00.26 | 00.23 | 00.21 |
| RN18-SRM | 00.48 | 00.63 | 01.82 | 01.29 | 00.51 |
| 10-Shot | | | | | |
| ACNN18 | 02.33 | 01.07 | 00.82 | 00.46 | 00.19 |
| ACNN18-KD | 00.39 | 00.35 | 01.47 | 00.53 | 00.50 |
| ACNN18-FitNet | 02.41 | 00.63 | 02.16 | 01.32 | 00.67 |
| ACNN18-AT | 00.69 | 00.35 | 02.90 | 00.40 | 01.76 |
| ACNN18-PKT | 00.53 | 01.92 | 01.18 | 00.59 | 00.26 |
| ACNN18-RKD | 01.10 | 00.50 | 00.25 | 00.76 | 00.57 |
| ACNN18-CRD | 01.13 | 00.25 | 00.77 | 00.85 | 02.98 |
| ACNN18-SRM | 00.58 | 01.39 | 04.33 | 00.47 | 01.84 |
| RN18 | 00.42 | 00.91 | 01.35 | 00.54 | 00.73 |
| RN18-KD | 00.33 | 00.53 | 01.13 | 00.78 | 00.91 |
| RN18-FitNet | 01.24 | 00.47 | 01.72 | 00.47 | 01.27 |
| RN18-AT | 00.42 | 00.55 | 02.72 | 00.38 | 01.41 |
| RN18-PKT | 00.52 | 02.56 | 00.63 | 00.62 | 00.54 |
| RN18-RKD | 00.28 | 00.14 | 00.71 | 00.34 | 00.27 |
| RN18-CRD | 00.05 | 00.13 | 00.66 | 00.49 | 01.67 |
| RN18-SRM | 01.06 | 01.86 | 06.84 | 01.21 | 02.95 |

Weight decay coefficient was set to $0.0001$. In addition, the pre-training phase took $80$ epochs with an initial learning rate of $0.1$, dropping by $0.1$ for every $20$ epochs.

References of the methods reported in Table 6 are:

- KD Hinton et al. (2015)
- AT Zagoruyko & Komodakis (2016)
- Seq. KD Yang et al. (2018)
- KD+ONE Zhu et al. (2018)
- ESKD Cho & Hariharan (2019)
- ESKD+AT Cho & Hariharan (2019)
- CRD Tian et al. (2020)

