# OpenReview forum: "Knowledge Distillation By Sparse Representation Matching"
_ICLR.cc/2021/Conference — Reject_

### Official Review · AnonReviewer4 · 2020-10-28
**Anonymous Review**

**Rating:** 3
**Confidence:** 4

**Review:**

<Paper summary>
This paper focuses on the problem of knowledge transfer between deep learning models, with the goal of transferring some of the information contained in a (typically large) teacher model to a smaller student network, improving performance of the latter. In particular, this work proposes to require the sparse representations of the activations of the teacher model to be 'similar' to that of the student model, as way of knowledge transfer. The method is presented and empirically evaluated.

<Review summary>
This reviewer likes the general motivation of this work, proposing that in some cases it might be better to enforce similarity between activations (or features) in a different domain, and requiring these representations to be sparse under some transformation is natural. While the general idea is appealing, several of the motivating claims are vague and the way these ideas are implemented (e.g. via classification to enforce similarity between vectors) are questionable.

<Details comments>
Strengths:
- the authors study an interest problem.
- the proposed method obtains good empirical performance.

Weaknesses:
- The idea of matching the representations for two different data in order to enforce some similarity (or in this context, 'knowledge transfer') has been extensively used. However, this only makes sense if the dictionaries for one and other case are related (see [1,2,3]). Enforcing the representations to be similar (or, as it's done here, to use the same leading atom) is reasonable when the atoms from one dictionary share some properties (or 'code' for related things) in the other.  In this work, the authors learn dictionaries for the teacher and student networks (Ds and Dt) completely independently, as there's no connection between the atoms in one and other dictionaries.
- The authors include an 'image-level labeling' which basically compares the mean value in the approximate features from the teacher network ($\tilde{t}$) to that of the vector of similarities of the student one ($k_s$). 1) I do not see how this is informative of relevant information between models, but more importantly 2) Comparing these two real numbers with a logistic loss makes little (if any) sense to me.
- The idea of employing sparse representations for data (in this case, the intermediate representations of networks) is natural. There is a large body of work that the authors seem to ignore. For example: on pg 2 they mention that "[sparse representations learning] were not proposed to be jointly optimized with other objectives". Please see refs [4-6] below for examples of this.
- Representations under redundant dictionaries are not unique, and the problem of finding sparse representations is NP-hard (see e.g [7]). Certainly, one can propose relaxations of this problem and even heuristic approximations (see e.g. [8]) but this is never discussed, and it is unclear how the obtained representations in this work fit in this context.

Smaller comments:
- The proposed method seems to be a pre-processing step: firs train teacher dictionaries, then student dictionaries and student weights, and then employ the KD method from Hinton et al. The authors should consider making this more explicit, perhaps detailing the full algorithm in a formal way.
- On page 4, the authors motivate the use of the sigmoid as activation function saying that the 'gradients in the backward pass are stable'. What does this mean? Would they be unstable if a ReLU was used instead (as used in most deep learning models and in the other models in this work)?
- At the end of Pixel-level labeling, stating the definition for $c_{n,i,j}$, the authors take the argmax over m, but there's no m in the expression.
- 'Relaxing' the problem of requiring the representations to be similar simply by turning a regression problem into a classification problem seems unfounded: why would the latter be easier than simply allowing the representations to be similar (say, with small L2 norm)?
- In defining the similarity kernel $\kappa$, I believe the $x$ and $y$ should be bold according to the authors notation.
- On a subjective note, the notation is not standard and thus a bit confusing: calligraphic capital letters usually denote sets or distributions, whereas here they denote vectors (as do bold non-calligraphic letters).

References:
1] Wang, Shenlong, et al. "Semi-coupled dictionary learning with applications to image super-resolution and photo-sketch synthesis." 2012 IEEE Conference on Computer Vision and Pattern Recognition. IEEE, 2012.

2] Qiu, Qiang, et al. "Domain adaptive dictionary learning." European Conference on Computer Vision. Springer, Berlin, Heidelberg, 2012.

3] Peleg, Tomer, and Michael Elad. "A statistical prediction model based on sparse representations for single image super-resolution." IEEE transactions on image processing 23.6 (2014): 2569-2582.

4] Mairal, Julien, Francis Bach, and Jean Ponce. "Task-driven dictionary learning." IEEE transactions on pattern analysis and machine intelligence 34.4 (2011): 791-804.

5] Sprechmann, Pablo, Alexander M. Bronstein, and Guillermo Sapiro. "Learning efficient sparse and low rank models." IEEE transactions on pattern analysis and machine intelligence 37.9 (2015): 1821-1833.

6] Monga, Vishal, Yuelong Li, and Yonina C. Eldar. "Algorithm unrolling: Interpretable, efficient deep learning for signal and image processing." arXiv preprint arXiv:1912.10557 (2019).

7] Mairal, Julien, Francis Bach, and Jean Ponce. "Sparse modeling for image and vision processing."

8] Makhzani, Alireza, and Brendan Frey. "K-sparse autoencoders." arXiv preprint arXiv:1312.5663 (2013).

---

> ### Author Response · Authors · 2020-11-17
> **Continued Reply to AnonReviewer4**
>
> **Comment 4**: *Representations under redundant dictionaries are not unique, and the problem of finding sparse representations is NP-hard (see e.g [7]). Certainly, one can propose relaxations of this problem and even heuristic approximations (see e.g. [8]) but this is never discussed, and it is unclear how the obtained representations in this work fit in this context.*
>
> **Reply**: Since we solve our sparse representation and learning objective using a standard stochastic optimizer, the solutions obtained are only locally optimal. Although analyzing the properties of the sparse representation learned in our method is an interesting research question and will give further insights into how to improve the sparse representations for the distillation objective, we believe that it is not necessary to demonstrate the main objective of our method. The current experimental results demonstrate that using a standard stochastic optimizer, we can achieve the objective of encoding and transferring knowledge from a teacher network to a student network and achieve very good performance.
>
> **Comment 5**: *The proposed method seems to be a pre-processing step: firs train teacher dictionaries, then student dictionaries and student weights, and then employ the KD method from Hinton et al. The authors should consider making this more explicit, perhaps detailing the full algorithm in a formal way.*
>
> **Reply**: We have added a summary of our algorithm at the end of Section 3
>
> **Comment 6**: *On page 4, the authors motivate the use of the sigmoid as activation function saying that the 'gradients in the backward pass are stable'. What does this mean? Would they be unstable if a ReLU was used instead (as used in most deep learning models and in the other models in this work)?*
>
> **Reply**: The sigmoid function used in computing similarities should not be considered as the activation function. In this context, it is considered as a kernel function that computes similarities between two points. Another popular choice for this purpose is the RBF kernel. In our preliminary tests we found that training using the RBF kernel is unstable as it exhibits numerical issues. The sigmoid kernel did not produce numerical issues, and this is why it was selected in our experiments. This is what we meant by using the word stable.
>
> **Comment 7**: *At the end of Pixel-level labeling, stating the definition for $c_{n,i,j}$, the authors take the argmax over m, but there's no m in the expression.*
>
> **Reply**: We have removed the “m” typo. Basically we are computing the index of the largest element in $t_{n,i,j}$ so the expression is simply argmax.
>
> **Comment 8**: *'Relaxing' the problem of requiring the representations to be similar simply by turning a regression problem into a classification problem seems unfounded: why would the latter be easier than simply allowing the representations to be similar (say, with small L2 norm)?*
>
> **Reply**: What we meant by this is that the task of learning the absolute representations for every pixel of every sample can be too restrictive and difficult, but learning how they are partitioned or classified into disjoint sets is a less restrictive and easier task. Since the latter does not enforce the absolute positions of samples in the sparse domain and only enforces that when two samples are “significantly encoded” by an atom (the most similar atom) in the dictionary of the teacher network, they should also be “significantly encoded” by one (the same for the two samples) atom in the dictionary of the student network. Obviously, there can be multiple solutions with different absolute representations that can satisfy our constraint, hence, it could be easier for the optimizer to solve.
>
> **Comment 9**: *In defining the similarity kernel $\kappa$, I believe the $x$ and $y$ should be bold according to the authors notation.*
>
> **Reply**: This is true and we have revised our manuscript to update this.
>
> **Comment 10**: On a subjective note, the notation is not standard and thus a bit confusing: calligraphic capital letters usually denote sets or distributions, whereas here they denote vectors (as do bold non-calligraphic letters).
>
> **Reply**: In our manuscript, we used calligraphic letters to denote the teacher network or student network. Vectors are denoted by bold-face small letters, while bold-face capital letters are used to denote matrices.

---

> > ### Comment · AnonReviewer4 · 2020-11-24
> > **Comments on responses**
> >
> > I thank the authors for their careful consideration of my comments. I have carefully read their comments and clarifications. Unfortunately, I'm still unconvinced that the work deserves greater merit than the one provided originally. Below are my response to the main comments.
> >
> > My main concern still regards the comment that "the dictionaries of the teacher network and student network are certainly dependent and related":
> > From the description in the paper and in the response, my understanding is that given the representation for the teacher's features, t, via a dictionary (D_T), a second dictionary (D_S) is trained so that the sparse representation computed via D_S on the features of the student model, k, use the same cardinality (i.e. same atom index) as the teacher's. The reason this still does not convince me is that there's nothing requiring t and k to be similar. As a result, if t differs from k, enforcing the representations \tilde{t} and \tilde{k} to use the j-th atom from their respective dictionaries D_T and D_S, does not imply anything about the relation between t and k. It is true that in their 'image-level labeling' the authors enforce some metric between \tilde{t} and k to be small, but this just requires the mean value over all pixels to be small, independent of where they are used. As a result, as far as I understand, \tilde{t} and \tilde{k} (for a given location) can be far from each other. Given that the weights of the networks are different, I do not see how \tilde{t} and \tilde{k} can be similar.
> >
> > -"What we meant by this sentence is that we have not seen any work that learns sparse representations with other objectives using deep neural networks and stochastic gradient descent"
> > This is also incorrect. Please see refs [9,10] below. What is correct (to the best of my knowledge) is its use for knowledge distillation in deep networks - thanks for the clarification of this in the text.
> >
> > - "Since we solve our sparse representation and learning objective using a standard stochastic optimizer, the solutions obtained are only locally optimal."
> > What do the authors mean by 'locally optimal' exactly? I agree that there's merit in demonstrating practical utility of a method. Instead, my comment referred to a lack of discussion of whether the solution of this sparse coding problem is unique. Such questions are important if one is to use these representations to evaluate 'similarity'.
> >
> > - "In our manuscript, we used calligraphic letters to denote the teacher network or student network. Vectors are denoted by bold-face small letters, while bold-face capital letters are used to denote matrices. "
> > This seems to be not true: the n-th input image is denoted by \mathcal{X}_n, and I'm assuming this is a an n-dimensional vector.. so shouldn't it be in bold (non-capital) letters?
> >
> > [9] "Learning fast approximations of sparse coding.", 2010.
> > [10] "On Multi-Layer Basis Pursuit, Efficient Algorithms and Convolutional Neural Networks", 2019.

---

> > > ### Author Response · Authors · 2020-11-24
> > > **Comments on the Responses**
> > >
> > > **Comment 1**: *My main concern still regards the comment that "the dictionaries of the teacher network and student network are certainly dependent and related": From the description in the paper and in the response, my understanding is that given the representation for the teacher's features, t, via a dictionary (D_T), a second dictionary (D_S) is trained so that the sparse representation computed via D_S on the features of the student model, k, use the same cardinality (i.e. same atom index) as the teacher's. The reason this still does not convince me is that there's nothing requiring t and k to be similar. As a result, if t differs from k, enforcing the representations \tilde{t} and \tilde{k} to use the j-th atom from their respective dictionaries D_T and D_S, does not imply anything about the relation between t and k. It is true that in their 'image-level labeling' the authors enforce some metric between \tilde{t} and k to be small, but this just requires the mean value over all pixels to be small, independent of where they are used. As a result, as far as I understand, \tilde{t} and \tilde{k} (for a given location) can be far from each other. Given that the weights of the networks are different, I do not see how \tilde{t} and \tilde{k} can be similar.*
> > >
> > > **Reply**:
> > >
> > > First of all, let us address the general argument in the main concern about dictionaries learned by our method being unrelated/independent. It is unclear to us what the Reviewer means exactly by using the expressions: “similar”, “far from each other”. We assume that in the Reviewer’s point of view, “similarity” here means that they are close in some proper metric. This is one way of enforcing similarities but it is not the only way. We have discussed in the manuscript that we do not want to impose an absolute similarity but a structural similarity. But more importantly, similarity is not a necessary condition for “dependency” or “relatedness” as the Reviewer claims that the dictionaries of the teacher and student network learned in our method are independent/unrelated.
> > >
> > >
> > > From the argument provided by the Reviewer, it seems that in the Reviewer’s opinion, absolute similarity is the only similarity as he/she says:
> > >
> > > *“As a result, as far as I understand, \tilde{t} and \tilde{k} (for a given location) can be far from each other. Given that the weights of the networks are different, I do not see how \tilde{t} and \tilde{k} can be similar.”*
> > >
> > > To see how tilde{t} and \tilde{k} (for a given location) can be far from each other ***but they can still be similar***, let us take the example from metric learning. The embeddings from the same class are not enforced to have similarities of 1 and their absolute similarity values can be very small. As long as these values are much larger than the similarity values computed with respect to embeddings from other classes, the structural or semantic similarity is preserved in the embedding space.
> > >
> > > That is, in our case, tilde{t} and \tilde{k} (for a given location) can be far from each other as long as for any position i and j, if tilde{t}_i and tilde{t}_j are encoded by the same atom, tilde{k}_i and tilde{k}_j are also encoded by one atom (the same for tilde{k}_i and tilde{k}_j).
> > >
> > > **Comment 2**: *Since we solve our sparse representation and learning objective using a standard stochastic optimizer, the solutions obtained are only locally optimal." What do the authors mean by 'locally optimal' exactly? I agree that there's merit in demonstrating practical utility of a method. Instead, my comment referred to a lack of discussion of whether the solution of this sparse coding problem is unique. Such questions are important if one is to use these representations to evaluate 'similarity'.*
> > >
> > >
> > > **Reply**: We agree that analyzing the uniqueness of the solutions obtained can provide extra merit to the paper but since our main objective is not about signal reconstruction and we do not enforce an absolute similarity constraint, the uniqueness of the sparse representation is not important as long as the sparse representation of the student network is structurally similar to the one discovered (by stochastic optimization) for the teacher.
> > >
> > > **Comment 3**: *In our manuscript, we used calligraphic letters to denote the teacher network or student network. Vectors are denoted by bold-face small letters, while bold-face capital letters are used to denote matrices. " This seems to be not true: the n-th input image is denoted by \mathcal{X}_n, and I'm assuming this is a an n-dimensional vector.. so shouldn't it be in bold (non-capital) letters?*
> > >
> > > **Reply**: An image is a higher-order tensor (not vector), which in tensor algebra, is often denoted by calligraphic letters. We agree that we forgot this point in the previous clarification.

---

> ### Author Response · Authors · 2020-11-17
> **Reply to AnonReviewer4**
>
> We would like to thank the Reviewer for spending time in reviewing our manuscript. Below is our reply to each comment.
>
> **Comment 1**: *The idea of matching the representations for two different data in order to enforce some similarity (or in this context, 'knowledge transfer') has been extensively used. However, this only makes sense if the dictionaries for one and other case are related (see [1,2,3]). Enforcing the representations to be similar (or, as it's done here, to use the same leading atom) is reasonable when the atoms from one dictionary share some properties (or 'code' for related things) in the other. In this work, the authors learn dictionaries for the teacher and student networks (Ds and Dt) completely independently, as there's no connection between the atoms in one and other dictionaries.*
>
> **Reply**: In our case, we are enforcing the representations of the same input signal to be similar. The representations come from different models (the teacher network and the student network), however, they are the representations of the same input signal. In addition, these representations are learned to solve the same learning objective (a classification task).
> More importantly, since the representations of the teacher network are used to generate labels, and thus, provide supervisory information to train the representations of the student, the dictionaries of the teacher network and student network are certainly dependent and related. It is unclear to us why they are not related and completely independent, as the reviewer argues.
>
> **Comment 2**:  *The authors include an 'image-level labeling' which basically compares the mean value in the approximate features from the teacher network ($\tilde{t}$) to that of the vector of similarities of the student one ($k_s$). 1) I do not see how this is informative of relevant information between models, but more importantly 2) Comparing these two real numbers with a logistic loss makes little (if any) sense to me.*
>
> **Reply**: The image-level labels are similar to the word histogram in the bag-of-features model. This widely used label or feature represents a global feature of an image in a single vector by aggregating the representations of its components (image patches or in our case pixels).
>
> The rationale behind using BCE loss is that since each element in this word histogram encodes “how much” a given atom is used to reconstruct the given input on a scale from 0 to 1, each element can be considered as a probability measure, and thus the BCE is used. This rationale is similar to the general practice in the image translation task (learning to transform one image to another) where the pixel intensity is considered as a probability measure and BCE is used as the loss function.
>
> **Comment 3**: *The idea of employing sparse representations for data (in this case, the intermediate representations of networks) is natural. There is a large body of work that the authors seem to ignore. For example: on pg 2 they mention that "[sparse representations learning] were not proposed to be jointly optimized with other objectives". Please see refs [4-6] below for examples of this.*
>
> **Reply**: We agree that our sentence is a bit misleading if it is seen as a generic statement. Since the sparse representation model is well studied and applied, it is natural to have been used with other objectives. What we meant by this sentence is that we have not seen any work that learns sparse representations with other objectives using deep neural networks and stochastic gradient descent. To avoid this confusion, we have removed this sentence and rewritten it as:
> “Although learning task-specific sparse representations has been proposed in prior works (Mairal et al., 2011; Sprechmann et al., 2015; Monga et al., 2019), we have not seen its utilization for knowledge transfer using deep neural networks and stochastic optimization”

---

### Official Review · AnonReviewer3 · 2020-10-28

**Rating:** 5
**Confidence:** 5

**Review:**

[Summary]
This paper proposes to do knowledge distillation via matching the coefficients of sparse representations. The sparse representation reconstructs deep neural networks’ intermediate representation via a set of over-complete dictionary and corresponding coefficients. The method matches the coefficients at two levels. One is at the pixel level, and the other is at the image level, which pools the coefficients from the pixel level. The experiments perform model compression for image classification tasks and show a performance gain over previous methods.

[Strengths]
1. The idea of using sparse representation is novel and interesting.
2. The clarity is good and easy to follow. The related works are sufficiently discussed.

[Weaknesses]
1. The method lacks some details, such as how D_T and D_S are learned and optimized. Please consider using an algorithm block to describe the full procedure of the proposed method.
2. The experiment part has a large room to be improved. It is known that KD methods are sensitive to the type of architectures used in the teacher or student model. Moreover, most of the methods are very sensitive to the choice of hyper-parameters. There are two concerns: (1) Does the proposed method generalize to a broader range of network architectures pairs? (ex: the analysis in Tian et al. 2020) (2) Do all methods in the table be given the same amount of tuning budget for the hyper-parameter? Based on the text, different methods are given a different tuning budget. For example, Tables 1 and 5 only have part of the methods that are tuned with a validation set while others use the default value from other papers, making the comparison unfair.
3. There are already more than a dozen distillation methods try to match the intermediate representation in a wide variety of ways, while most of them seem to work. Maybe matching a random projection of intermediate representation between teacher and student will work as well. Then why we want to use a more complicated method? The work will be more valuable if it provides a more in-depth insight into why using sparse representation helps. The paper has some arguments in the introduction, but they are not convincing since the coefficients can be the same even though the dictionaries between teacher and student are very different. A mathematical explanation is preferred.

---

> ### Author Response · Authors · 2020-11-17
> **Continued Reply to AnonReviewer3**
>
> **Comment 3**: *There are already more than a dozen distillation methods try to match the intermediate representation in a wide variety of ways, while most of them seem to work. Maybe matching a random projection of intermediate representation between teacher and student will work as well.
> Then why we want to use a more complicated method? The work will be more valuable if it provides a more in-depth insight into why using sparse representation helps. The paper has some arguments in the introduction, but they are not convincing since the coefficients can be the same even though the dictionaries between teacher and student are very different. A mathematical explanation is preferred.*
>
> **Reply**: We are not aware of a method using random projections for knowledge distillation. We would appreciate to get a reference to a specific paper describing such an approach.
> The mathematical and practical motivation of using sparse representation has been well studied in the field of Signal Processing, e.g. see the reference in our paper to the work of Zhang et al. 2015. In the context of KD, the basic observation that motivates the use of sparse representations is that neighboring pixels are often highly correlated, thus, compressible and possess sparse representation in some domains (in Signal Processing for example a commonly used domain for such methods is the Fourrier domain or the Wavelet domain). Such a compressible representation is better at capturing essential information in the signal and has been utilized in other applications. Our work shows that compressible representations are also suitable for representing knowledge in a KD method.
> We think that research on improving the training of smaller models using KD methods plays an important role in the effort of making the use of deep networks practical, especially due to the fact that the current trend in achieving state-of-the-art performance in various problems almost exclusively requires the use of deep networks with ever increasing number of parameters

---

> ### Author Response · Authors · 2020-11-17
> **Reply to AnonReviewer3**
>
> We would like to thank the Reviewer for spending time in reviewing our manuscript. Below is our reply to each comment.
>
> **Comment 1**: *The method lacks some details, such as how D_T and D_S are learned and optimized. Please consider using an algorithm block to describe the full procedure of the proposed method.*
>
> **Reply**: We have added a summary of our algorithm at the end of Section 3 in the revised manuscript.
>
> **Comment 2**: *The experiment part has a large room to be improved. It is known that KD methods are sensitive to the type of architectures used in the teacher or student model. Moreover, most of the methods are very sensitive to the choice of hyper-parameters. There are two concerns: (1) Does the proposed method generalize to a broader range of network architectures pairs? (ex: the analysis in Tian et al. 2020) (2) Do all methods in the table be given the same amount of tuning budget for the hyper-parameter? Based on the text, different methods are given a different tuning budget. For example, Tables 1 and 5 only have part of the methods that are tuned with a validation set while others use the default value from other papers, making the comparison unfair.*
>
> **Reply**: In relation to the first concern of this comment, we agree with the reviewer that KD methods can be sensitive to the type of architectures used.
>
> It is always desirable to include in the experimental analysis as many network architectures as possible. However, we had to strive for a balance between i) comparing the proposed method with existing methods and ii) analysing different aspects of the proposed method, when designing our experiments because of limited resources. For this reason, we compared all methods using two representative architectures (a network architecture with residual connections and another one without residual connections) in the transfer learning tasks.
>
> In terms of evaluating the generalization performance of the competing methods, the ImageNet dataset is the golden standard dataset for image classification and, thus, the good empirical results of the proposed method in ImageNet do indicate the strengths of our method. The code of the paper is publicly available (submitted as supplementary material with this ICLR submission) and can be used by anyone to evaluate our method with other architecture pairs.
>
> In relation to the second concern, we should note that the standard practice (also used in Tian et al. 2020 mentioned by the Reviewer) is to adopt the best hyperparameter values suggested by the original paper proposing the competing methods. This approach is followed for a good reason: in most cases, it is impractical to tune exhaustively all hyperparameter values for all competing methods, and thus, due to that the hyper-parameter values suggested in the original paper of a method have been carefully determined, those values are commonly used in the experiments on the same datasets. The exhaustive search is needed in order to also determine a good range of hyperparameter values (and possible combinations of values for multiple hyperparameters of some methods) to be included in the computational budget of each method, thus impractical. For this reason, we believe that the protocol used in our paper, as in other KD papers, e.g., in Tian et al. 2020, is reasonable.

---

### Official Review · AnonReviewer1 · 2020-10-28
**The method presented looks promising but cleaner experiments are required to clarify its working elements and provide comparison to a close aompetitor**

**Rating:** 5
**Confidence:** 3

**Review:**

Strength and weaknesses:
+
The method is reasonable and competitive. Especially for the task transfer scenario is seems to advance the state of the art

-
Comparison is lacking as a main relevant competitor was not tested
I believe the method can be much simplified by instead of using sparse decomposition using plain clustering of pixel columns. Experiments testing this option should be conducted
The task transfer experiments, whose results are the most impressive, are done without weight Initialization of the student networks. It is not clear why, and the results may be different if initialized networks are used.
At the bottom line: cleaner experiments (adding comparison to Jain et al, using plain clustering instead of sparse decomposition, using student networks with ImageNet initialized weights) are required to take this paper beyond reasonable doubt.


Detailed comments:

-	Page 2: “However, we argue that the intermediate feature maps by themselves are not a good representation of the knowledge encoded in the teacher to teach the students” – This sentence states a claim, which is a main claim of this paper. However, the claim is not supported by any argument or justification in the introduction
-	Page 3: the subscript \Tau in D_{\Tau}^{(l)} seems to be meaningless (why do we need it?)
	Later is becomes clearer as D_S is introduced, but a note should be given before to make \Tau meaningful
-	Page 4: the pixel-labels are based on the 1-nn dictionary item, and so the knowledge transfer proposed in actually based a simple clustering of the pixels (each pixel represented by a single cluster index), not on the sparse representation. First, this means that K=1 can be naturally used, as K>1 is not really adding information to be transferred. Second, the method is very similar to transfer by clustering, seemed to be proposed by (Jain et al.) (based on the description of this work in the relevant work – I am not familiar with it)
-	Page 4: the image labels are enforcing similarity of word histograms between teacher and student. It is not clear, tough, why the BCE loss is used and not some histogram distance loss (l_1, Xi square, or even l_2). BCE is appropriate when the entrees in the two input vectors are probabilities of binary classification for independent problems. This is not the case here.
-	Page 5: it seems that comparison to the most similar method of (Jain et al.,) is not conducted
-	Page 6: the experiments showing that SRM is not sensitive to \mu and \lambda increase the likelihood that simple clustering, where each pixel is hard-assigned to one of the clusters can be used instead of sparse code encoding (we know that the identity of the single first dictionary item is the only one transferred in the pixel-labels, so it is likely that similarity to other dictionary items is not relevant)
-	The task transfer results (table 5) are the most impressive results of the paper, indicating clear advantage of SRM over competitors. However, two drawbacks:
o	Why are the student networks randomly initialized and not initialized with their trained ImageNet weights (at least for ResNet 18 weights are publically available, I believe). Using pre-trained weight is likely to improve accuracy, and it is not clear why they are avoided
o	Comaprison to Jain et al is missing
-	Section 4.3: the teacher accuracy is missing

---

> ### Author Response · Authors · 2020-11-17
> **Continued Reply to AnonReviewer1**
>
> **Comment 6**: *Page 6: the experiments showing that SRM is not sensitive to \mu and \lambda increase the likelihood that simple clustering, where each pixel is hard-assigned to one of the clusters can be used instead of sparse code encoding (we know that the identity of the single first dictionary item is the only one transferred in the pixel-labels, so it is likely that similarity to other dictionary items is not relevant)*
>
> **Reply**: As we have pointed out in the third comment, the size of the dictionary and the sparsity degree can have an effect on the learned sparse representation, thus affecting the identity of the nearest atom (pixel-level labels) as well as the image-level label. While we showed that our method is not very sensitive to \mu and \lambda, this does not indicate that using only pixel-level labels suffices (the results with and without image-level label indicate the necessity of image-level label in our method), let alone using simple clustering for the pixel-level label.
>
> **Comment 7**: *The task transfer results (table 5) are the most impressive results of the paper, indicating clear advantage of SRM over competitors. However, two drawbacks: o Why are the student networks randomly initialized and not initialized with their trained ImageNet weights (at least for ResNet 18 weights are publically available, I believe). Using pre-trained weight is likely to improve accuracy, and it is not clear why they are avoided o Comaprison to Jain et al is missing*
>
> **Reply**: We did not initialize the student with pretrained ImageNet weights because of two reasons:
> - In real-life scenarios, we might not have access to the data that has been used to train the pretrained models.
> - To train the student network on pretrained data might be too computationally expensive.
>
> Besides, when the student is randomly initialized, the performance of the transfer learning method shows the ability of a given method to transfer both the pretrained knowledge and the task-specific knowledge.
>
> Finally, this evaluation protocol is standard practice and was also used in other knowledge distillation papers, for example in Passalis et al (ECCV 2018), Park et al (CVPR 2019) and so on.
>
> **Comment 8**: *Section 4.3: the teacher accuracy is missing*
>
> **Reply**: We have included the teacher’s accuracy in the revised manuscript for completeness.

---

> ### Author Response · Authors · 2020-11-17
> **Reply to AnonReviewer1**
>
> We would like to thank the Reviewer for spending time in reviewing our manuscript. Below is our reply to each comment.
>
> **Comment 1**: *Page 2: “However, we argue that the intermediate feature maps by themselves are not a good representation of the knowledge encoded in the teacher to teach the students” – This sentence states a claim, which is a main claim of this paper. However, the claim is not supported by any argument or justification in the introduction.*
>
> **Reply**: The paragraph that follows our argument in the introduction explains our motivation to use sparse representation: sparsity is a well-studied and widely used representation model. When the signal exhibits certain properties, e.g. local smoothness, it possesses a sparse representation in some domain. More specifically, in our case, neighboring pixels in hidden feature maps of CNN are highly correlated. This observation motivates the usage of sparse representation.
>
> **Comment 2**: *Page 3: the subscript \Tau in D_{\Tau}^{(l)} seems to be meaningless (why do we need it?) Later is becomes clearer as D_S is introduced, but a note should be given before to make \Tau meaningful*
>
> **Reply**: The subscript \mathcal{T} is used to denote the teacher model, as opposed to the subscript \mathcal{S}, which denotes the student model. We have added this clarification in the revised manuscript.
>
> **Comment 3**: *Page 4: the pixel-labels are based on the 1-nn dictionary item, and so the knowledge transfer proposed in actually based a simple clustering of the pixels (each pixel represented by a single cluster index), not on the sparse representation. First, this means that K=1 can be naturally used, as K>1 is not really adding information to be transferred. Second, the method is very similar to transfer by clustering, seemed to be proposed by (Jain et al.) (based on the description of this work in the relevant work – I am not familiar with it)*
>
> **Reply**: Regarding the first point in this comment, the value of K can have an effect on the learned sparse representation and thus the nearest atom since we are optimizing a reconstruction objective, not the clustering objective. Besides, we also take advantage of the sparse representation to compute the image-level label, thus the sparse representation and hence the value of K can affect the result. In the experiment section, we included a comparison with and without using this image-level label. The results indicate that using the image-level label can improve the distillation performance.
> For the second point in this comment, in the Related Works section, we have discussed the similarities and differences between our work and Jain et al.
>
> **Comment 4**: *Page 4: the image labels are enforcing similarity of word histograms between teacher and student. It is not clear, tough, why the BCE loss is used and not some histogram distance loss (l_1, Xi square, or even l_2). BCE is appropriate when the entrees in the two input vectors are probabilities of binary classification for independent problems. This is not the case here.*
>
> **Reply**: The rationale behind using BCE loss is that since each element in this word histogram encodes “how much” a given atom is used to reconstruct the given input on a scale from 0 to 1, each element can be considered as a probability measure, and thus the BCE is used. This rationale is similar to the general practice in the image translation task (learning to transform one image to another) where the pixel intensity is considered as a probability measure and BCE is used as the loss function.
>
> **Comment 5**: *it seems that comparison to the most similar method of (Jain et al.,) is not conducted*
>
> **Reply**: The method of Jain et al indeed has some similarities with our method but there are many differences as pointed out in the introduction. Since there was no publicly available implementation of this method, hence the lack of certain details, we could not include this method in our comparison. Besides, by using only the pixel-level label and excluding the image-level label, our method and the method of Jain et al are similar in the sense that both learn to transfer the partition of intermediate representations, and we showed that solely learn to transfer this partition is inferior to our method.

---

> ### Comment · AnonReviewer1 · 2020-11-23
> **I have read the rebuttal. While some of my comments did got replies I do not change my general jusgement**
>
> The main issue which concerns me is that the sparse coding was not compared to simple clustering and I 'm not convinced it is required. Currently, I think simple column clustering may be able to  achieve the same effect (if used to transfer knowledge  using the same techniques: pixel wise and image wise).
> Hence the main point of the paper is not clear enough empirically.

---

> > ### Author Response · Authors · 2020-11-23
> > **Using simple clustering, there is no image-level label**
> >
> > Here we should stress that our method works using both pixel-level labels and image-level labels. That is, the idea of deriving pixel-level (local) labels and image-level (global) labels is part of the novelty of the work. We have shown in our experiments that using the image-level labels (computed from sparse representation) is important. If we only use a clustering method for deriving the pixel-level labels, how can we derive the image-level labels?

---

> > > ### Comment · AnonReviewer1 · 2020-11-23
> > > **As far as I can see, there is no problem to define image-level labels with plain clustering**
> > >
> > > You can use the image-level labels exactly as you do with the sparse representation: by constructing histograms of cluster (codeword) activity for the teacher and the student and demanding them to be similar.
> > > Essentailly, you do not have to change anything from what you currently do in this respect. Just apply the spatial aggregation (the sum over i,j in equation 5 to the cluster posterior probability tensor in which T_{i,j,k}=P(h=k| position i,j), i.e. the probability that cluster k codeword is present at position (i,j).
> > > My belief is that plain clustering will do the same as sparse coding while being much simpler. I may be wrong, but you should show this empirically.

---

> > > > ### Author Response · Authors · 2020-11-23
> > > > **Proper comparison requires thorough investigation of the variant suggested by the Reviewer**
> > > >
> > > > What the Reviewer suggests might seem simpler at the first sight but to properly execute the idea requires many considerations.
> > > >
> > > > For example, given that we have the labels generated by a clustering method as the Reviewer suggested, it is not straight-forward how to optimize the student network with this label. Should we alternate between optimizing the student’s parameters using SGD and clusters’ centroids using the clustering method to compute the pixel-level and image-level predictions generated by the student? If it is the case, then we need to decide after how many epochs of SGD that we will perform clustering to define the centroids.
> > > >
> > > > What kind of loss function should we use for the image-level label? In our case, we use BCE loss but this requires the values in the image-level label to fall into the range [0,1]. That is, how to compute the probability that a given input belongs to a given cluster? (for K-means we only compute the membership based on euclidean distance).
> > > >
> > > > Any variant to define the pixel-level label and image-level labels requires proper experimentation in order to conclude whether it works or not. Even if some ideas may seem simpler than the proposed method (as the one mentioned by the Reviewer) selection of a specific idea requires a large number of follow-up choices which are needed to make the idea work, those follow-up choices being neither simple nor well-motivated.
> > > >
> > > > Regarding the notion of “simplicity”, the proposed method can be easily implemented in a few lines of code by using standard functionalities provided by all DL frameworks in the form of a neural layer/module, while the implementation of the approach proposed by the Reviewer is more complex in terms of implementation, as it requires the implementation of efficient (online) and effective (robust in terms of initialization) online clustering (especially important for big datasets like ImageNet for which we have to compute similarities in a mini-batch manner), a functionality that to the best of our knowledge is not provided by DL frameworks. Thus, the measurement of the simplicity of the two approaches (ours and the one mentioned by the Reviewer) is not as simple as mentioned by the Reviewer.
> > > >
> > > > Since the choices made for formulating the method in our paper come from a well-justified motivation (sparse representation model) and we have demonstrated that it works under different experiment settings, we believe inspecting an extension of our idea is unnecessary and out of the scope of this work.

---

> > > > > ### Comment · AnonReviewer1 · 2020-11-24
> > > > > **I think most of the issues raised w.r.t clustering have simple answers**
> > > > >
> > > > > Answering briefly to the points raised:
> > > > > - The clustering should be trained as part of the network layer, with SGD. This is simple to implement for Kmeans or GMM, with several standard layers and tensor operatrions
> > > > > - Soft Kmeans computes (the kind it is easy to implement in a net) computes P(h=i|X)  - the probability for class i in the location. Summed across the spaital dimensions, and notmalize by HW (just like your equation) it provides a normalized histogram with values in [0,1]. I do not see the issue
> > > > > - Indeed, empiric science requires experiments and tuning - nothing new about this. This is not a drawback
> > > > > - Simplicity: Using a single cluster index (or plain probability of cluetr presence) is a simpler idea than sparse coding conceptually. It is also not harder to implement. Ocam's razor tells us that the second (sparse coding) should be used only if it has a clear advantage over the former
> > > > > - "sparse coding is a " well-justified motivation": sparse coding it s respectable tradition, but it is not justified in the current context if a simpler idea can do the same work

---

> > > > > > ### Author Response · Authors · 2020-11-24
> > > > > > **We do not agree with the argument of the Reviewer**
> > > > > >
> > > > > > We do not agree with the Reviewer on the notion of conceptual simplicity as this is subjective and we believe the concept of sparse representation is as easy to understand and simple as a clustering concept. But more importantly, there is no argument against the usefulness of our method. There is an alternative suggestion that may or may not work, while we show that our method works very well.
> > > > > >
> > > > > > Finally, we agree with the Reviewer that Sparse Coding is a respectable tradition, and we never claimed otherwise. Since sparse representation is a respectable tradition for compressible signals (which holds in our context with neighboring pixels being highly correlated), the use of sparse representation is well-motivated and effective as shown in our experiments while the usefulness of the idea suggested by the Reviewer is a conjecture.

---

### Official Review · AnonReviewer2 · 2020-10-29
**Review of "Knowledge Distillation By Sparse Representation Matching"**

**Rating:** 4
**Confidence:** 5

**Review:**

**Paper summary**
This paper proposes a knowledge distillation on the feature maps using sparse representation. The proposed method firstly constructs an over-complement dictionary to express the teacher's feature maps and learn sparse representation to express the teacher's feature map using the dictionary. Since directly utilizing the sparse representation is a too strong restriction for the student network, the loss function is designed to find the indices of sparse codes. The proposed distillation method is validated through several experiments.

**Pros**
1. This paper proposes a way of utilizing sparse representation for knowledge distillation.
2. The algorithm is written in clear formulations.

**Cons**
1. The main idea of sparse representation matching (SRM) is the combination of sparse representation and knowledge distillation. However, the actual implementation of SRM does not transfer the sparse representation of the teacher to the student. Only the indices or the entire image's sparse code were transferred via knowledge distillation, so the feature map information of the teacher is not transferred. This looks counter-intuitive and weakens the arguments of the paper. Therefore, it is necessary to describe the information transferred by SRM in detail and the rationale for using this kind of information transferring.
2. All experiments were conducted on All-CNN, but this network is not usually used by other knowledge distillation papers, so the experiments should be re-conducted on a more standard and efficient setting. For example, All-CNN got 74.7% accuracy using SRM with 2.2M parameters (in table 1), but a recent paper [1] (CRD) got 75.5% accuracy using WRN16-2 with 0.7M parameters, which uses only 1/3 number of parameters. In other words, All-CNN is not proper to compare with other distillation methods. Use popular network architectures (WRN, ResNet, VGG etc.) to get more reasonable performance.
[1] Contrastive Representation Distillation (ICLR 2020)
3. In the case of transfer learning, it is common to tune the learning rates and weight decays for each dataset and each network. However, the experiments in the paper consistently use the same learning hyper-parameters. As a result, the performance gap between the baseline and the proposed method seems to have greatly inflated. A great gap against baseline (about 10%~20%) seems to be very different from the results of the other distillation papers.  In short, to be fair comparisons with other distillation methods, learning parameter tuning is necessary.

---

> ### Author Response · Authors · 2020-11-17
> **Reply to AnonReviewer2**
>
> We would like to thank the Reviewer for spending time in reviewing our manuscript. Below is our reply to each comment.
>
> **Comment 1**: *The main idea of sparse representation matching (SRM) is the combination of sparse representation and knowledge distillation. However, the actual implementation of SRM does not transfer the sparse representation of the teacher to the student. Only the indices or the entire image's sparse code were transferred via knowledge distillation, so the feature map information of the teacher is not transferred. This looks counter-intuitive and weakens the arguments of the paper. Therefore, it is necessary to describe the information transferred by SRM in detail and the rationale for using this kind of information transferring.*
>
> **Reply**: Our first rationale is that enforcing the student network to directly learn to regress to the same sparse representation produced by the teacher is a too restrictive task since this involves learning the absolute value of every point in a high-dimensional space. On the other hand, learning the structure of the sparse domain is less restrictive in the sense that we only enforce the relative constraints between samples in the feature space. In this sense, our method is similar to previous methods such as Relational Knowledge Distillation (PKD) (Park et al, 2019) or Contrastive Knowledge Distillation (CRD) (Tian et al, 2020). In RKD or CRD, relative distances between samples are used as the relative constraint while in our method, the partition of the sparse domain by the codebook is used as the relative constraint
> Section 3.2 in the revised manuscript has been updated to reflect this point.
>
> **Comment 2**: *All experiments were conducted on All-CNN, but this network is not usually used by other knowledge distillation papers, so the experiments should be re-conducted on a more standard and efficient setting. For example, All-CNN got 74.7% accuracy using SRM with 2.2M parameters (in table 1), but a recent paper [1] (CRD) got 75.5% accuracy using WRN16-2 with 0.7M parameters, which uses only 1/3 number of parameters. In other words, All-CNN is not proper to compare with other distillation methods. Use popular network architectures (WRN, ResNet, VGG etc.) to get more reasonable performance. [1] Contrastive Representation Distillation (ICLR 2020)*
>
> **Reply**: We should highlight that our experiments do contain results with ResNet18 as the student network in Transfer Learning and ImageNet experiments. The All-CNN architecture is a simple feed-forward architecture and is very similar to VGG net. Here we should note that our focus is on the comparison between distillation methods given the same student and teacher, rather than finding the best student architecture. Since our experiments contain results for both residual (ResNet18) and non-residual (All-CNN) networks on many datasets, including the golden standard ImageNet dataset, we believe the results do reflect our objective of comparisons.
>
> **Comment 3**: *In the case of transfer learning, it is common to tune the learning rates and weight decays for each dataset and each network. However, the experiments in the paper consistently use the same learning hyper-parameters. As a result, the performance gap between the baseline and the proposed method seems to have greatly inflated. A great gap against baseline (about 10%~20%) seems to be very different from the results of the other distillation papers. In short, to be fair comparisons with other distillation methods, learning parameter tuning is necessary.*
>
> **Reply**: In almost all knowledge distillation papers in the references, the general practice is to keep the same learning rate schedule and weight decay for all algorithms. Thus, we believe that our protocol is aligned with the current practice of the field when comparing knowledge distillation methods.

---

### Decision · Program_Chairs · 2021-01-07
**Final Decision**

**Decision:**

Reject

**Comment:**

The paper received four negative reviews. The overall idea was found to be interesting, but several concerns were raised. There is a general consensus that the experimental part and the results are not convincing. Several comments have also been made regarding the clarity and motivation, which needs to be strengthened. R4 also mentions references from the sparse estimation literature that would help for positioning the paper. The rebuttal did address some of these points, but it was not sufficient to change their opinion.

Overall, the area chair agrees with the reviewers and follows their recommendation.